# Bringing weak transitions to light

Yu He ®[1]✉, Xiao-Min Tong ®[2], Shuyuan Hu[1], Gergana D. Borisova ®[1], Hao Liang ®[3], Maximilian Hartmann ®[1], Veit Stooß[1], Chunhai Lyu ®[1], Zoltán Harman[1], Christoph H. Keitel ®[1], Kenneth J. Schafer ®[4], Mette B. Gaarde ®[4], Christian Ott ®[1]✉ & Thomas Pfeifer ®[1]✉

Weak transitions between quantum states are of fundamental importance for a broad range of phenomena from analytical biochemistry to precision physics, but generally challenge experimental detection. Due to their small cross sections scaling with the absolute square of their transition matrix elements, spectroscopic measurements often fail in particular in the presence of competing background processes. Here we introduce a general concept to break this scaling law and enhance the transition probability by exploiting a stronger laser-coupled pathway to the same excited state. We demonstrate the concept experimentally by attosecond transient absorption spectroscopy in helium atoms. The quasi-forbidden transitions from the ground state $1s^2$ to the weakly coupled doubly excited $2p3d$ and $sp_{2,4-}$ states are boosted by an order of magnitude. Enhancing single-photon-suppressed transitions can find widespread applicability, from spectral diagnostics of complex molecules in life and chemical sciences to precision spectroscopy of weak transitions in metastable atomic nuclei in the search for new physics.

Transitions between quantum states underlie modern spectroscopy and are integral to a variety of disciplines. Weak transitions are of particular relevance to optical clocks[1–6] and fundamental-physics experiments[7]. Even dipole-allowed transitions can be very weak: In the first theoretical interpretation of the seminal observation of doubly excited states in helium using synchrotron radiation[8], the unobserved transition from the ground state to the $sp_{2,n-}$ series was described as quasi-forbidden[9]. The even weaker dipole-allowed transition to the $2pnd$ series was first experimentally resolved about 30 years later[10]. According to Fermi's golden rule[11], the absorption cross section scales with the absolute square of the transition matrix element in traditional atomic and molecular spectroscopy performed with low-intensity laser light. Breaking this scaling law could boost the direct experimental detection of weak transitions.

The well-known optical theorem[12]

$$\sigma \propto \mathrm{Im}(A) \tag{1}$$

relates the total cross section $\sigma$ to the forward scattering amplitude $A$. For absorption near a resonance, the response function in the linear regime leads to $A \propto |T|^2$, with $T$ being the complex-valued transition matrix element. In the presence of intense light at different frequencies, the response function modifies to

$$\tilde{A} \propto T^*(T + T') , \tag{2}$$

where $T'$ accounts for the contribution of additional pathways beyond the direct single-photon excitation. For weak direct transitions, $T'$ can be much larger than $T$, enabling a significant enhancement of the spectral visibility of weak transitions.

The key idea of our method is illustrated in Fig. 1, based on a few-level scheme. The interaction of the considered system with the weak broadband laser 1 results in a coherent excitation from the ground state $|g\rangle$ to two excited states $|1\rangle$ and $|2\rangle$. We consider the state $|1\rangle$ to be very weakly coupled to the ground state compared to state $|2\rangle$ owing to the small transition matrix element $|T_{g1}| \ll |T_{g2}|$. Its faint spectral signal [Fig. 1b] is easily lost or buried in noise. If state $|1\rangle$ can be strongly

[1]Max-Planck-Institut für Kernphysik, Heidelberg, Germany. [2]Center for Computational Sciences, University of Tsukuba, Ibaraki, Japan. [3]Max-Planck-Institut für Physik komplexer Systeme, Dresden, Germany. [4]Department of Physics and Astronomy, Louisiana State University, Baton Rouge, Louisiana, USA. ✉e-mail: yuhe@mpi-hd.mpg.de; christian.ott@mpi-hd.mpg.de; thomas.pfeifer@mpi-hd.mpg.de

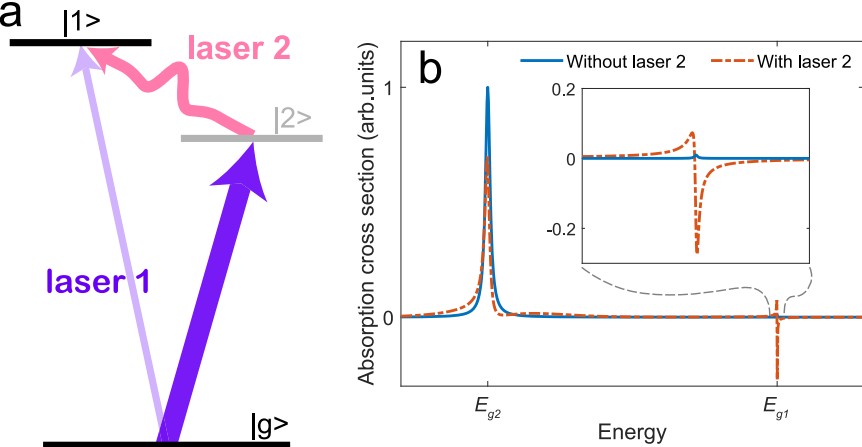

**Fig. 1 | Schematic illustration of the concept of enhancing spectroscopically weak transitions. a** Laser 1 creates a coherent superposition of ground and excited states, with state $|1\rangle$ being very weakly excited compared to state $|2\rangle$ as a result of a vanishingly small transition matrix element $|T_{g1}| \ll |T_{g2}|$. Laser 2 opens an additional pathway by coupling state $|1\rangle$ to the more strongly populated excited state $|2\rangle$. The resulting spectral signal on the originally weak transition $|g\rangle \rightarrow |1\rangle$ gets enhanced as shown in panel (**b**). Parameters for the calculation results shown in panel (**b**) are

as follows: state $|1\rangle$ ($E_{g1}$ = 62 eV, 400-fs decay time); state $|2\rangle$ ($E_{g2}$ = 60 eV, 17-fs decay time); transition matrix elements for $T_{g1}$, $T_{g2}$, and $T_{21}$ are 0.0008, 0.04, and 2 a.u., respectively; laser 1 [0.2-fs FWHM (full width at half maximum) duration, 9.1-eV spectral bandwidth, $1 \times 10^{11}$-W/cm² peak intensity, 60 eV central frequency] overlaps temporally with laser 2 (4-fs FWHM duration, $2 \times 10^{12}$-W/cm² peak intensity, 700 nm central wavelength).

coupled to state $|2\rangle$ by an additional (intense) laser, as illustrated in Fig. 1a, the resulting transfer of quantum-state amplitude from $|2\rangle$ to $|1\rangle$ will lead to an enhanced and more detectable spectral signal as shown in Fig. 1b.

## Results

### Experimental demonstration

To experimentally demonstrate this concept, we turn to the helium atom and its $2p3d$ and $sp_{2,4-}$ ($^1P^o$) doubly excited states, whose transition probabilities from the ground state $1s^2$ ($^1S^e$) are a few orders of magnitude lower than that of the $2s2p$ ($^1P^o$) state (see Supplementary Table 1). Throughout this work, we followed the early notation of the $^1P^o$ doubly excited states below the $N$ = 2 threshold of He$^+$ first introduced by Cooper, Fano, and Prats[9]: $sp_{2,n\pm}$ (the strong mixing of $2snp$ and $2pns$) and $2pnd$, in which the electron configurations are from the shell-model picture (e.g., $2snp$ implies that one electron is in the $2s$ state and the other in the $np$ state). For more classification schemes of these states, see[13,14]. However, both the $2p3d$ and the $sp_{2,4-}$ states can be strongly coupled to the $2s2p$ state with two visible (VIS) photons, via the intermediate $2p^2$ ($^1S^e$) state as shown in the relevant energy-level scheme in Fig. 2b. This additional coupling pathway opens the possibility of enhancing the spectral visibility of these two weak resonances. A common beam-path transient absorption beamline[15] is utilized in the experiment as depicted in Fig. 2a, in which the few-cycle VIS pulses propagate collinearly with the extreme-ultraviolet (XUV) pulses produced by high-harmonic generation in neon. The ultrashort, few-cycle VIS pulse allows the transfer of quantum-state amplitude to happen within the short lifetime of the $2s2p$ state. A variable time delay between the VIS and XUV pulses is introduced by a piezo-driven split mirror. Both pulses are focused into a helium-filled target gas cell with a backing pressure of ~200 mbar. The transmitted XUV radiation is dispersed by a grating and detected by a CCD camera. The absorption spectrum is characterized by the optical density $OD(\omega, \tau) = -\log_{10}[I(\omega, \tau)/I_0(\omega)]$, where $I(\omega, \tau)$ and $I_0(\omega)$ are the transmitted and incident XUV spectra, respectively. $\omega$ is the frequency, and $\tau$ denotes the time delay between the two pulses. The delay convention is such that a positive time delay means that the target helium is first excited by the weak XUV pulse and then exposed to the stronger VIS pulse. The spectral resolution of the spectrometer is 40 meV near 63.7 eV, determined by fitting the spectral profile of $sp_{2,3+}$ with a

natural linewidth of 8.3 meV[16]. The sub-20 meV energy spacing between the $2p3d$ and $sp_{2,4-}$ states makes them indistinguishable in our measured spectroscopic data.

When only the weak XUV pulse is present, the absorption spectrum shown in Fig. 2c agrees with the reported measurements performed with synchrotron radiation[10,16–18], with the spectral features of the $sp_{2,n+}$ series being predominant and the $sp_{2,n-}$ series barely visible. When the additional VIS pulse is employed, the spectral profiles of the $sp_{2,n+}$ series are modified as shown in Fig. 2c. The variation of the strongly coupled states of the $sp_{2,n+}$ series was studied earlier and enabled the extraction of amplitude and phase modifications[19] and the reconstruction of a correlated two-electron wave packet[20] from modified Fano absorption line shapes[21].

We now focus on the spectral feature around the weakly coupled $2p3d$ and $sp_{2,4-}$ resonances at ~64.1 eV, the visibility of which is much enhanced in the presence of the VIS laser pulse. While hardly visible in the XUV-only case, the additional VIS laser turns their relative spectral amplitude comparable to the neighboring lines. We note that these neighboring lines belonging to the $sp_{2,n+}$ series have much larger couplings to the ground state compared to the $2p3d$ and $sp_{2,4-}$ states (see Supplementary Table 1). Figure 3a (upper panel) shows the transient absorption spectrum in the vicinity of these two resonances. The spectral profile is barely visible for negative time delays, then starts to become very pronounced in the pulse-overlap region, and gradually fades away with increasing time delay. Furthermore, it exhibits a significant spectral dip below the background absorption by the action of the VIS pulse arriving later in time [lower panel of Fig. 3a], which persists for positive time delays and shows oscillations in amplitude. The observed spectral profile thus resembles window-type resonances, while neither the $sp_{2,4-}$ nor the $2p3d$ states naturally exhibit such line shapes.

### Theoretical modeling

To model our experimental observations, we performed a large-scale numerical simulation by solving the two-electron time-dependent Schrödinger equation (TDSE) using the hyperspherical close-coupling method[22]. We use an XUV pulse with a full width at half maximum (FWHM) duration of 0.2 fs and a peak intensity of $1 \times 10^{11}$ W/cm², centered at 60 eV. The VIS pulse has a central wavelength of 700 nm, a FWHM duration of 4 fs, and a peak intensity of $2 \times 10^{12}$ W/cm². The generalized photoabsorption cross section[23,24] is calculated, which can

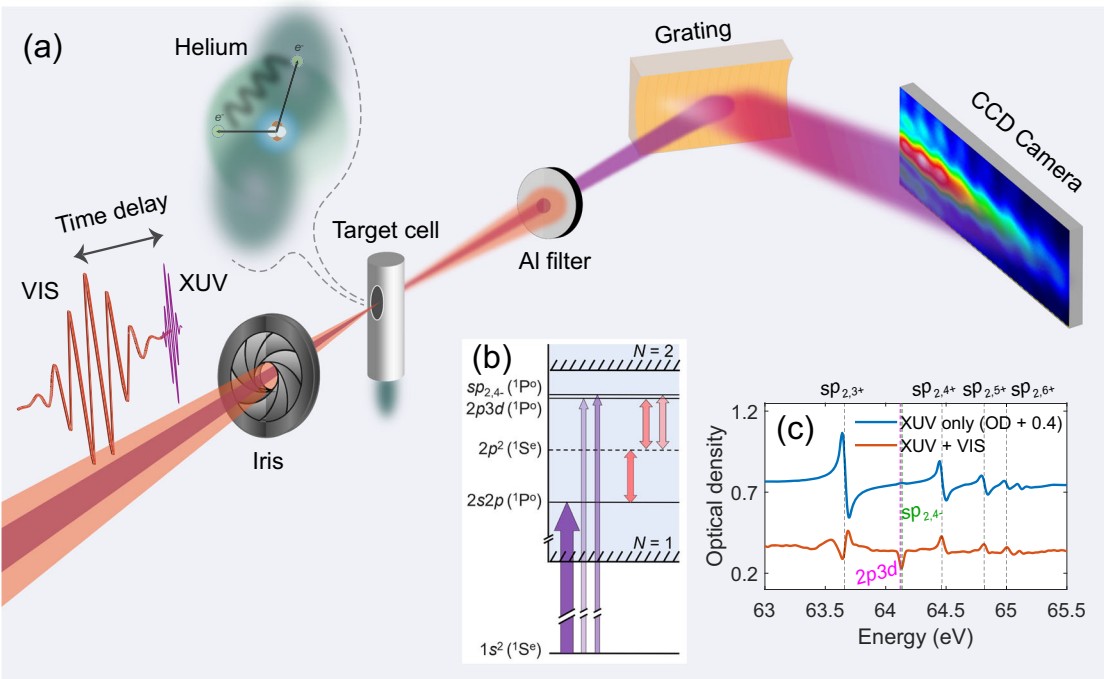

**Fig. 2 | Outline of the attosecond transient absorption measurement. a** Illustration of the experimental setup. CCD, charge-coupled device. **b** Relevant energy-level diagram in helium. **c** Absorption profile at 1.5-fs time delay (reddish-orange line) along with the result in the absence of the VIS pulse (blue line).

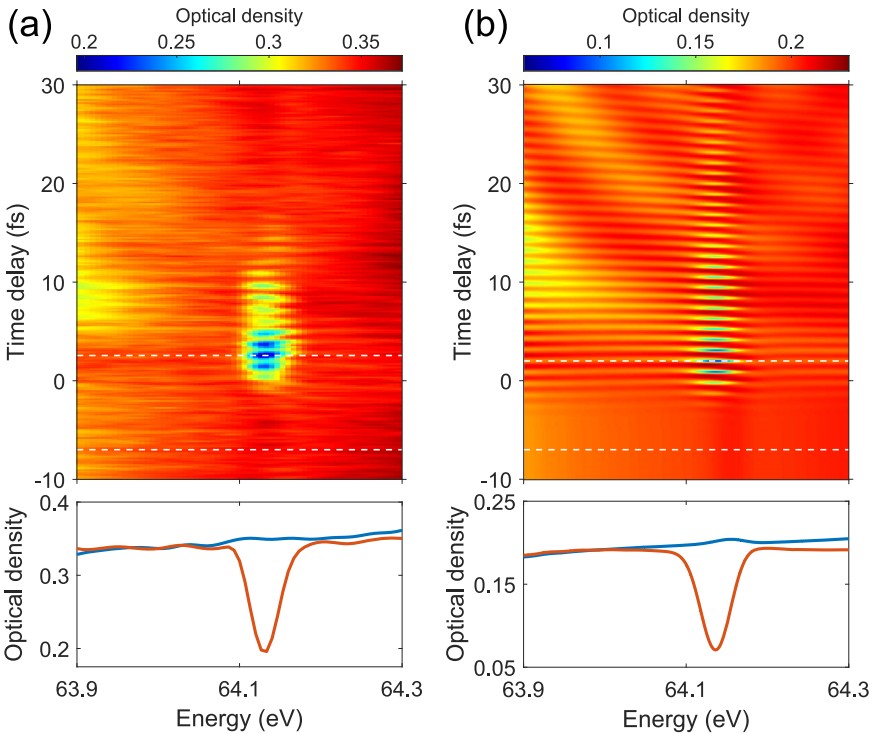

**Fig. 3 | Experimental and theoretical demonstration. a** Upper panel: measured transient absorption spectrum in the vicinity of the $2p3d$ and $sp_{2,4-}$ states; lower panel: absorption profiles at two representative time delays [blue (reddish-orange) line: negative (positive) time delay] marked as white dashed lines in the upper panel. **b** Simulated spectrum and absorption profiles after implementing the spectral broadening and the spectrometer resolution.

be both positive and negative and reflects the photon-energy dependent loss or gain for a helium atom interacting with the two-color field. Compared to the case of large negative time delays which corresponds to effectively XUV-only interaction, the generalized absorption cross section for both $2p3d$ and $sp_{2,4-}$ states gets enhanced by an order of magnitude (see Supplementary Fig. 1). In order to compare with the

measured results shown in Fig. 3a, one needs to include the spectral broadening and the finite spectrometer resolution (see 'Data processing' in Methods). The simulated transient absorption spectrum and its lineouts after this procedure are shown in Fig. 3b, which agree well with the experimental profiles and the dynamical evolution with time delay. The lower contrast of the oscillation observed in the experiment

is explained by experimental time-delay jitter and spatial beam inhomogeneity. The $2s2p \to 2p^2 \to 2p3d/sp_{2,4-}$ two-VIS-photon coupling is identified to be the dominant pathway for the enhanced spectral visibility and its temporal behavior, indicated by the 1.1-fs beating period, which corresponds to the energy difference between $2s2p$ and $2p3d/sp_{2,4-}$ states of ~3.9 eV. Its resonant enhancement is further confirmed by the calculation through artificial removal of the $2p^2$ state (see Supplementary Fig. 2). The finite time-delay window for the observation of the spectral enhancement is linked to the ~17-fs lifetime of the $2s2p$ state[25,26], which serves as the initial state of the two-VIS-photon coupling pathway. We note that due to our experimental grazing-incidence geometry[15], the walk-off of the focus of the XUV beam leads to less spatial overlap with the VIS beam in the interaction region at large time delays. This effect gives rise to a faster decay of the experimental absorption signal than the theory results. The intersecting yellowish features shown in the transient absorption spectra in Fig. 3 are hyperbolic sidebands of the $sp_{2,3+}$ state, which converge to its energy position with increasing time delay and result from its perturbed free-induction decay[24,27,28].

## Discussion

In summary, we have presented and experimentally verified a general concept to enhance the spectroscopic signal of weak transitions by exploiting a stronger laser-coupled pathway. As an experimental demonstration, we report the first time-resolved observation of the weakly XUV-populated $2p3d$ and $sp_{2,4-}$ ($^1P^o$) doubly excited states in helium by attosecond transient absorption spectroscopy. With tunable coupling lasers, more weakly coupled autoionizing states are expected to be observed and studied by ultrafast time-resolved spectroscopy in helium and other atomic and molecular species, enabling the tracking and coherent control of correlated multielectron dynamics in the time domain. Similar to other well-established methods like cavity-enhanced spectroscopy[29] and surface-enhanced infrared spectroscopy[30], enhancing single-photon-suppressed transitions in matter can find technological applications in analytical spectroscopy and spectral fingerprinting with enhanced sensitivity in life, chemical, and material sciences. Moreover, with the aid of intermediate states[31] or bridge processes[32], this concept holds substantial potential for fundamental physics: We envision it to enable identification and efficient driving of extremely weakly coupled states (including nuclear states) of relevance to future atomic clocks and precision spectroscopy for fundamental-physics tests in searches for physics beyond the standard model[1-7,33].

## Methods

### Transition matrix elements

To compute the transition matrix elements for the doubly excited states, we developed a program to solve the two-electron Schrödinger equation for helium. A similar strategy can be found in the work by Feist[34]. In short, the angular degree of freedom of the two-electron wave function $\Psi(\mathbf{r}_1, \mathbf{r}_2)$ is expanded in a coupled angular basis:

$$\Psi(\mathbf{r}_1, \mathbf{r}_2) = \sum_{L,M,l_1,l_2} f^{L,M}_{l_1,l_2}(r_1, r_2)|l_1 l_2 LM\rangle, \quad (3)$$

with total angular momentum $L$ and magnetic number $M$. The radial coordinates are discretized with a finite-element discrete-variable representation[35]. According to R-matrix theory[36], the continuum wave function $|1sEl\rangle$ is the solution to a linear equation:

$$(H - E - E_{1s})|1sEl\rangle = |1s\rangle_1\delta(r_2 - r_{\max}) + \delta(r_1 - r_{\max})|1s\rangle_2, \quad (4)$$

where $H$ is the Hamiltonian of the He atom. A minimal residual method[37] is applied to solve this linear equation. The doubly excited states are approximately represented by the continuum states with the same real part of energy, normalized in a finite box. With the wave

function, the transition dipole matrix elements could be computed directly as listed in Supplementary Table 1.

### Experimental apparatus

The commercial laser system (FEMTOPOWER™ HE/HR CEP4) delivers 20 fs, 3 mJ pulses with a central wavelength of 780 nm at 3 kHz repetition rate. In order to broaden the spectrum, the laser pulses are guided through a 1.5 m-long double differentially pumped hollow-core fiber, which is filled with helium with a gas pressure of ~2 bar. The pulses are temporally compressed by 7 pairs of chirped mirrors (Ultrafast Innovations PC70) and are characterized by a dispersion-scan (D-scan) setup. The D-scan measurement yields a FWHM pulse duration of 4.6 fs. Afterwards, the few-cycle VIS laser pulses (725 nm central wavelength, ~1 mJ pulse energy) are sent into the vacuum beamline[15] to perform the transient absorption measurements. The XUV pulses are produced by high-harmonic generation (HHG) in a cell containing neon gas with a backing pressure of 100 mbar. To account for the plasma-induced blue shift introduced in the HHG process, a VIS pulse with a 700 nm central wavelength and 4-fs FWHM duration is employed in the numerical simulation.

### Data processing

In order to compare with the experimental results, the spectral broadening and the finite spectrometer resolution have to be taken into account in the simulation results, especially for the narrow resonances as considered here. To this end, we first convolute the calculated generalized absorption cross section $\sigma(\omega)$ as shown in Supplementary Fig. 1 with a normalized Gaussian function $G(\omega, \Delta\omega_1)$ of FWHM width $\Delta\omega_1$ to account for the Doppler broadening (the pressure broadening is much smaller and is hence ignored)

$$\sigma_D(\omega) = \sigma(\omega) * G(\omega, \Delta\omega_1). \quad (5)$$

Regarding the consideration of spectrometer resolution, we would like to point out that the commonly used approach by directly convolving the cross section with a Gaussian function is inappropriate, since what is recorded by the spectrometer in the experiment is the transmitted laser spectrum. After obtaining the Doppler-broadened cross section $\sigma_D(\omega)$ from Eq. (5), one should apply the convolution to the transmitted laser spectrum before computing the optical density OD, which takes the following form after applying the linear Beer-Lambert law

$$OD(\omega) = -\log_{10}\frac{[I_0(\omega)e^{-\sigma_D(\omega)\rho l} * G(\omega, \Delta\omega_2)]}{I_0(\omega)}. \quad (6)$$

Here, one assumes the target medium to be homogeneous and the cross section $\sigma_D$ space-independent. However, it has been shown that the temporal reshaping of the driving pulse resulting from resonant pulse propagation effects inside the medium leads to reduced absorption compared to what is predicted by the Beer-Lambert law[38]. To account for this effect and reach a comparable enhancement of resonant OD relative to the background as in the experiment, we used a pathlength-density product $\rho l = 3.6 \times 10^{17}$ cm$^{-2}$ rather than $7.7 \times 10^{17}$ cm$^{-2}$ retrieved from the experiment. This explains the smaller background OD shown in Fig. 3b as compared to that in Fig. 3a. For the simulation results shown in the main text, the Doppler broadening $\Delta\omega_1 = 400$ μeV and the spectrometer resolution $\Delta\omega_2 = 40$ meV are employed.

According to the Beer-Lambert law, the transmitted laser spectrum will be exponentially damped or amplified depending on the sign of the cross section $\sigma_D$. Due to the convolution with the laser spectrum in Eq. (6), absorption features in the transmitted spectrum are highly suppressed as compared to the emission, which underlies the persistent window-like resonant feature in the absorption spectrum despite the comparable amplitudes of positive and negative absorption cross sections.

## Data availability

The data for the figures have been deposited in Zenodo at https://zenodo.org/records/15507771.

## Code availability

The codes supporting the findings of this study are available from the authors upon request.

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

## Acknowledgements

We thank Jan-Michael Rost for helpful discussions. X.-M.T. was supported by the Multidisciplinary Cooperative Research Program in the Center for Computational Sciences, University of Tsukuba, and by Grants-in-Aid for Scientific Research (Grants No. JP22K03493) from the Japan Society for the Promotion of Science. The work at LSU was supported by the U.S. Department of Energy, Office of Science, Basic Energy Sciences under Contract No. DE-SC0010431.

## Author contributions

Y.H., S.H., G.D.B., M.H., and V.S. performed the attosecond transient absorption experiment. X.-M.T. performed the TDSE calculations. H.L. and C.L. calculated the transition matrix elements. Y.H. did the data analysis and developed the concept, with input from X.-M.T., S.H., G.D.B., H.L., M.H., V.S., C.L., Z.H., C.H.K., K.J.S., M.B.G., C.O., and T.P. Y.H., C.O., and T.P. wrote major parts of the manuscript. All authors contributed to the discussions of the results and the writing of the manuscript.

## Funding

## Competing interests

The authors declare no competing interests.
