## [Transparent Peer Review file · Nature Communications]

Bringing Weak Transitions to Light

Corresponding Author: Dr Yu He

Version 0:

Reviewer comments:

Reviewer #1

(Remarks to the Author)
see attached PDF.

Reviewer #2

(Remarks to the Author)

The paper discusses the challenge of detecting weak transitions between quantum states due to their small cross sections. The authors introduce a method to enhance these transitions by using a stronger laser-coupled pathway to the same excited state. They demonstrate this concept experimentally with attosecond transition absorption spectroscopy in helium atoms, significantly boosting quasi-forbidden transitions. This enhancement technique has potential applications in various fields, including spectral diagnostics of complex molecules and precision spectroscopy of weak transitions in metastable atomic nuclei.

I recommend the article for publication in Communications Physics as the claims made by the authors are clearly articulated and supported by both experimental observations and accurate numerical calculations, which are in agreement with one another. As mentioned in the abstract, this topic may be of interest to researchers beyond the specific application to helium. My only minor concern is that articles in Nature Communications are intended for a broad range of researchers who may not be familiar with the specific types of two-electron levels utilized in this work. While some readers may be comfortable with configurations such as $2p3d$, they may not be as familiar with states such as $sp2,4-$. To ensure the article is accessible to a wider audience, it could be beneficial to provide additional explanations or context for these and similar terms.

Otherwise, I support the publication of this article in Nature Communications.

Reviewer #3

(Remarks to the Author)

This manuscript describes and demonstrates a very interesting concept for enhancing weak transitions in transient absorption spectroscopy. The idea is to exploit the coupling of the weak transitions to strong transitions by a laser field. The concept is demonstrated on the $2p3d$ and $sp2,4-$ states of helium, the transitions from the ground state to which are nearly forbidden. By coupling them to the strong $2s2p$ transition, the weak transitions are significantly enhanced. Additionally, the intensity of the weak transition is modulated with a period of 1.1 fs, corresponding to the 3.9-eV separation between the $2s2p$ and the $2p3d/sp2,4-$ states.

Overall, this is a very good manuscript presenting convincing experiments and theory. I recommend the following revisions:

1) The authors should comment on the time profiles of the absorption line shown in Fig. 3. Whereas the calculations appear to be in line with the mentioned 17-fs decay of the $2s2p$ state, the experimental absorption signal appears to decay much faster and in a markedly non-exponential fashion, maybe even a time-dependent shift.

2) It would be desirable to see a larger portion of the experimental data than shown in Fig. 3a. A presentation of the

experimental data as in Ext. Data Fig. 1 would be very helpful to better visualize the experimental results.

3) The authors mention the broad applicability of their concept to other types of spectroscopy. The mentioned applications remain somewhat vague, which could be significantly improved by mentioning specific cases to which the present concept could be applied.

4) Additional details on the experimental parameters should be given to facilitate the reproduction of the data. For example, how were the VIS pulses compressed, how was their duration measured, where was the cutoff of the XUV continuum, etc.

5) The numerical parameters used for the calculations shown in Fig. 1 should also be given.

Overall, this is a very good manuscript, which should be publishable in Nat. Comm. following these revisions.

Version 1:

Reviewer comments:

Reviewer #1

(Remarks to the Author)

The authors have carefully and positively considered all the questions I have raised in my review report. As all the answers to these questions are correct and proper changes have been made in the revised version of the manuscript, I can now fully endorse publication of this version of the manuscript in Nature Communications.

Reviewer #3

(Remarks to the Author)

I am satisfied with the revisions and replies, such that I recommend publication of this manuscript in its present form.

Responses to reviewers' comments

Please find below our detailed reply to the referees' reports and the corresponding changes we made in the manuscript. In this response letter, referees' remarks appear in *italicized blue*, our responses to them are in standard black and the changes made to the manuscript and Supplemental information are shown in **red**.

Reviewer #1 (Remarks to the Author):

Referee-Report-Yu-He-NatComm

Title: "Bringing Weak Transitions to Light," by Yu He et al.

Summary:

A joint theoretical and experimental study of the purely correlated process of double excitation in helium atoms is demonstrated using time-delayed XUV attosecond pulse and visible light. Here, the authors employ a visible light to strongly couple doubly-excited states (DESS) and report on a large enhancement of some weak transitions. For negative time delays, a coherent superposition of several DESS with $^1P^o$ symmetry is created thanks to the broad bandwidth of the weak XUV attopulse centered at 60 eV (near on resonance with the $2s2p$ DES). Among those states, clearly resolved spectrally are the spectral features of the $sp_{2,n+}$ series and not the $2pnd$ series and $sp_{2,n-}$ series, as there is a large contrast between their transition probabilities from the ground state. It is only when a few-cycle visible light is present (positive time delays) that the spectral profiles of the $sp_{2,n+}$ series and $sp_{2,n-}$ series are found to be modified. Focusing on the $2p3d$ and $sp_{2,4-}$ states (separated only by sub-20 meV), with each being strongly coupled to the $2s2p$ state by two-photon transition via the $2p^2$ ($^1S^e$) DES, the authors reported on an

enhancement of one order of magnitude for those two states, and that their spectral profile bears close resemblance with window-type resonances.

Results/claims:

The authors report on the first time-resolved observation in the vicinity of the weakly XUV-populated $2p3d$ and $sp_{2,4}$ - doubly-excited states in helium atoms by ATAS. Although the oscillator strengths of those states are much weaker than that for the $2s2p$ state, their calculations correctly showed that the transition probabilities from the $2s2p$ ($1P^o$) state to the $2p2$ ($1S^e$) state or from the latter state to the $2p3d/sp_{2,4}$ - state are comparable. Therefore, as the energy gap between these states of opposite different symmetry quite matches the wavelength (700 nm) of the tunable visible light, the authors demonstrated that the two-photon coupling between the $2s2p$ state and each of these two states with the same symmetry mediated by a few-cycle visible light brings those two states (initially buried in noise) to light.

Assessment:

Enhancing single-photon-suppressed transitions, as discussed in this contribution, underlie modern spectroscopy. The concept of introducing another field to induce additional couplings and thus break the scaling law, to enhance some weak transitions is of relevance to optical clocks and fundamental-physics experiments. However, this concept is certainly not new, see for instance [22-24]. Even for helium atoms of interest here, it has already been demonstrated how modified the spectral profiles of strongly coupled states of the $sp_{2,n+}$ series are. The novelty of this work is that it focuses on one state from the $2pnd$ series and $sp_{2,n-}$ series (whose transition probability from the ground state is much weaker than that for the $sp_{2,n+}$ series) normally buried in noise, and is able to bring them to light (i.e., to enhance their weak transition).

The experimental technique (ATAS) and the theoretical method (TDSE approach within the hyperspherical close-coupling method) employed here are well-established tools in

the AMO community. It is only when effects of the spectral broadening and finite spectrometer resolution are included in the theory that theory and experiment agree quite well. Thus, the numerical method is sound. Technically, the paper is well-structured and clearly written. The figures are carefully designed. The analysis of the results is compelling and supports the conclusions drawn by the authors. However, there are still many aspects regarding the physics and control of this effect that are not elaborated in the main text. I have seven major and some technical issues that the authors should address before I can make a decision on whether the physics discussed in the revised manuscript can be published in Nature Communications.

Response: We wish to express our gratitude to the referee for their detailed reading and the comprehensive assessment of our manuscript. We have carefully addressed the concerns below and made a proper revision of the manuscript. We believe that the substantial improvements to the manuscript and the clarification of all raised points could enable referee 1 to join the other two referees in recommending our work for publication.

Major points

1. Discrepancy between theory and experiment: In Figure 3, the disagreement between theory and experiment at longer time delays is attributed to the experimental time-delay jitter and spatial beam inhomogeneity. For the absorption profiles at the two representative time delays, why do the optical densities from theory and experiment differ by a factor roughly of two?

Response: The linear Beer-Lambert law is applied when simulating the optical density from the calculated absorption cross section, which assumes that the target medium is homogeneous and the cross section is space-independent. This assumption remains valid when the target medium is optically thin and the temporal reshaping of the XUV pulse due to pulse propagation effects inside the medium is insignificant. However, it has been shown that pulse propagation effects lead to reduced absorption compared to what

is predicted by the Beer-Lambert law [M. D. Crisp, Phys. Rev. A **1**, 1604 (1970)]. Since transmitted spectrum (and hence the OD) is sensitive to the exponent (σ_D and ρl) in Eq. (6), in our data processing, we used a pathlength-density product of $3.6 \times 10^{17} \text{ cm}^{-2}$ (rather than $7.7 \times 10^{17} \text{ cm}^{-2}$ retrieved from the experiment) to account for this effect and reach a comparable enhancement of resonant optical density relative to the background as in the experiment. This leads to a smaller background OD in the simulation results. We have commented on it in the revised edition of the manuscript accordingly.

In Data processing, after introducing the linear Beer-Lambert law, we added: **Here, one assumes the target medium to be homogeneous and the cross section σ_D space-independent. However, it has been shown that the temporal reshaping of the driving pulse resulting from pulse propagation effects inside the medium leads to reduced absorption compared to what is predicted by the Beer-Lambert law³⁸. To account for this effect and reach a comparable enhancement of resonant OD relative to the background as in the experiment, we used a pathlength-density product $\rho l = 3.6 \times 10^{17} \text{ cm}^{-2}$ rather than $7.7 \times 10^{17} \text{ cm}^{-2}$ retrieved from the experiment. This explains the smaller background OD shown in Fig. 3b as compared to that in Fig. 3a. (Page 11)**

Fig. Same as Fig. 3b in the main manuscript except for a different pathlength-density product of $7.7 \times 10^{17} \text{ cm}^{-2}$ used in the data processing.

For reference, the figure above shows the calculated transient absorption spectrum and the lineouts when $\rho l = 7.7 \times 10^{17} \text{ cm}^{-2}$ is used. One clearly sees the large negative OD values and the much more significant spectral enhancement as compared to the experimental observation.

2. Presence of the $2p^2$ ($1S^e$) state in the absorption profile in Figure 2(c): The enhancement mechanism proposed the authors involves the $2p^2$ ($1S^e$) contribution as an intermediate state. If this is correct, then this state should be directly visible in the experimental absorption profile in Fig. 2(c) at lower energy. I urge the authors to show that figure for energy starting from at least 59 eV.

Response: We included the transient absorption spectrum in a large energy range of 58-66 eV in the Supplementary information (Supplementary Figure 3). However, no resonant feature appears in the vicinity of $2p^2$ ($1S^e$) at ~62 eV due to the dipole selection rule since it has the same (even) parity as the ground state $1s^2$ ($1S^e$) (thus the transition is electric dipole forbidden).

Supplementary Figure 3. **Measured transient absorption spectrum in a large energy range.** Contrary to the spectral evolution of $2p3d/sp_{2,+}$ states shown in Fig. 3 in the main manuscript, the spectral amplitude of $2s2p$ state drops significantly in the pulse-overlap region, and gradually rises with increasing time delay. This is related to the loss of amplitude of $2s2p$ state by the action of the coupling VIS pulse when it arrives at the helium target after the XUV pulse. (SI, page 4)

3. Modification of the 2s2p feature: *In the new figure 2(c) above, how does the intensity for the 2s2p signal vary when the visible light is on or off? Given the nature of the coupling at that fixed visible pulse intensity, was it expected?*

Response: In the newly added Supplementary Figure 3 (shown above), one can see that the spectral amplitude of the 2s2p state drops significantly during the pulse overlapping region, and gradually rises with increasing time delay. This is contrary to the evolution of 2p3d and sp_{2,4} states and results from the expected loss of amplitude of the 2s2p state when the coupling VIS pulse is present. This observation also agrees with the theory calculations shown in the Supplementary Figure 1a.

In the caption of Supplementary Figure 3, we added: **Contrary to the spectral evolution of 2p3d/sp_{2,4} states shown in Fig. 3 in the main manuscript, the spectral amplitude of 2s2p state drops significantly in the pulse-overlap region, and then gradually rises with increasing time delay. This is related to the loss of amplitude of 2s2p state by the action of the coupling VIS pulse when it arrives at the target helium after the XUV pulse.** (SI, page 4)

4. Effects of the visible light intensity: *How do the enhancement of the 2p3d and sp_{2,4}-states together with the intensities of the 2s2p and 2p² (¹S^e) signals change as the visible light intensity increases? What is the minimal and maximal visible intensities for which this effect occurs?*

Response: For moderately intense VIS pulses, the described spectral enhancement of 2p3d and sp_{2,4} states gets more pronounced with increasing VIS pulse intensity, as more quantum-state amplitude can be transferred to them. However, since these two states are close to the N=2 ionization threshold of 65.4 eV, the ionization loss [mechanism discussed in A. Kaldun *et al.*, Science 354, 738 (2016), V. Stooß *et al.*, Phys. Rev. Research 2, 032041 (2020)] at even higher intensities for both the 2s2p and 2p3d/sp_{2,4} states will result in decreased spectral enhancement. Based on the measurements shown

below (which we included in the Supplementary information as **Supplementary Figure 4**), we observe pronounced spectral enhancement in the VIS intensity range of $\sim 1 - 7$ TW/cm².

Supplementary Figure 4. Effects of the VIS intensity on the spectral enhancement of $2p3d/sp_{2,4}$ states. The spectral visibility first increases with increasing VIS intensity since more quantum-state amplitude can be transferred to these two states, which subsequently decreases due to the ionization loss of the $2s2p$ and $2p3d/sp_{2,4}$ states at high VIS intensities. Pronounced spectral enhancement is shown in an energy range of 1-7 TW/cm². (SI, page 5)

The transient absorption spectra in a large energy range for different VIS intensities are shown below. No resonant feature appears in the vicinity of $2p^2$ ($1S^e$) at ~ 62 eV. For the $2s2p$ state, its spectral feature is only slightly modified for low VIS intensity. Its spectral amplitude drops for high VIS intensities at small positive time delays due to the loss of state amplitude (because of coupling to other states and ionization loss), which rises gradually with increasing positive time delay. These results are not included in the manuscript since they are not the main focus of this work and might distract potential readers.

Fig. Transient absorption spectra in a large energy range for different VIS intensities.

5. Effects of the visible light CEP: *Given the few-cycle nature of the visible light, can the enhancement be manipulated by varying the carrier-envelope phase of the visible light?*

Response: For different CEP values, the carrier electric field of the visible light becomes different. Nevertheless, only slight differences in amplitude appear in the calculation results for different CEPs shown below, while the general spectral evolution with respect to time delay is pretty robust and the oscillations are still in phase. In the experiment, because of the intrinsic phase locking of the attosecond pulses to the half-cycles of the generating intense visible light [Paul *et al.*, Science **292**, 1689 (2001)], these delay-dependent oscillations remain even when the CEP is not stabilized. We included the calculation results in the Supplementary information as Supplementary Figure 5.

Supplementary Figure 5. **Effects of the carrier-envelope phase (CEP) of the VIS pulse.** Differences in resonant OD values are observed by varying the VIS light CEP, while the general spectral evolution with respect to time delay is pretty robust and the oscillations are still in phase. (SI, page 6)

6. Quantum beats: Quantum beats between the $2s2s$ and $2p3d/sp2,4-$ states are briefly mentioned on page 8, before Conclusions. Fringes are observed in the upper panels of both Figure 3(a) and 3(b). For both experiment and theory, I urge the authors to display a comparative time-delay dependence of the optical density for three values of the energy: 64 eV, 64.15 eV, 64.2 eV. Take the time delay to be less than the $2s2p$ lifetime.

Response: The optical density as a function of time delay is shown below for three different energies (64 eV, 64.2 eV, and the energy with maximum OD enhancement). Compared with the theory calculations shown in the panel **b**, the experimental results in panel **a** show a lower contrast of oscillation, which is explained by experimental time-delay jitter and spatial beam inhomogeneity. We included this figure as **Supplementary Figure 6** in the Supplementary information.

Supplementary Figure 6. **OD lineouts as a function of time delay at three different energy positions.** Except for a lower contrast of the oscillation observed in the experiment results in panel **a**, which is explained by experimental time-delay jitter and spatial beam inhomogeneity, the observed features are well captured by the theory calculations in panel **b**. (SI, page 6)

7. Intersecting yellowish features in figure 3: Both theory and experiment in the upper panels of Figure 3 show yellowish intersecting features, which unfortunately are not elaborated in the manuscript. I urge the authors to do so. Are they related to the lifetimes of the doubly-excited states involved in the coupling?

Response: We thank the referee for pointing out these interesting intersecting yellowish features in Figure 3. They are called hyperbolic sidebands in related transient absorption studies [M. Wu *et al.*, JPB **49**, 062003 (2016), J. Rørstad *et al.*, PRA **96**, 013430 (2017)] and are characteristic features of the perturbed free-induction decay [M. Lindberg and S. W. Koch, PRB **38**, 7607 (1988)]. These sidebands are determined by hyperbolic curves $(E - E_r) \times \tau = \text{constant}$, which converge to the resonant energy position E_r with increasing time delay τ . In our case, they originate from the neighboring $sp_{2,3+}$ state (slightly lower in energy than $2p3d/sp_{2,4-}$ states at ~ 63.7 eV).

In the revised manuscript, we added: **The intersecting yellowish features shown in the transient absorption spectra in Fig. 3 are hyperbolic sidebands of the $sp_{2,3+}$ state, which converge to its energy position with increasing time delay and result from its perturbed free-induction decay^{24,27,28}.** (Page 8)

Minor points

1. Fermi's Golden rule: The authors claimed in the first paragraph of the introduction that the absorption cross section is proportional to the square of the transition amplitude. Isn't the square modulus? Is the transition matrix element a real or complex number?

Response: In general, the transition matrix element is a complex number and it should be the square modulus in Fermi's Golden rule. We thank the referee for pointing it out and we have revised it accordingly:

In the abstract: Due to their small cross sections scaling with the **absolute** square of their transition matrix elements, spectroscopic measurements...

In the first paragraph: According to Fermi's golden rule, the absorption cross section scales with the **absolute** square of the transition matrix element in traditional...

In the second paragraph: For absorption near a resonance, the response function in the linear regime leads to $A \propto |T|^2$, with T being the **complex-valued** transition matrix element.

*2. Equation (2): This claim is translated in Equation (2), which would be incorrect if the transition amplitude is complex. In this case, I would expect this equation to be: $|T|^2 + 2\text{Re}[T^*T']$ granted that $|T|^2$ can be neglected.*

Response: Considering the complex-valued transition matrix elements, Equation (2) should be written as $\tilde{A} \propto T^*(T + T')$. We have made this revision accordingly.

The reason is as follows: The probability of finding the atom in the excited state $|c_e|^2$, is related to, but different from, the probabilities in absorbing the XUV photons. This can be understood from the Maxwell-Bloch equation, where the probability (A) in absorption/emission of photon is proportional to (the dipole moment of the atom T^*) \times (the off-diagonal element $\rho_{ge} = c_g c_e^* \approx c_e^*$). Here, we take $c_g = 1$ as we are in the weak-field regime. This expression indicates that it is not the population of the excited state $|c_e|^2$, but its state coefficient c_e , that determines the light absorption/emission.

When only the weak XUV laser is present, one has $c_e^* \propto T$. Therefore, one gets the usual relation $A \propto |T|^2$ in the linear regime. However, when there are different pathways beyond the single-XUV-photon excitation to reach the excited state, the expression of its state coefficient modifies to $c_e^* = T + T_1 + T_2 + \dots$. Although the probability in finding the atom in the excited state is given by $|c_e|^2 = |T + T_1 + T_2 + \dots|^2$, the photoabsorption cross section is related to $T^* c_e^* = T^*(T + T_1 + T_2 + \dots)$.

Reviewer #2 (Remarks to the Author):

The paper discusses the challenge of detecting weak transitions between quantum states due to their small cross sections. The authors introduce a method to enhance these transitions by using a stronger laser-coupled pathway to the same excited state. They demonstrate this concept experimentally with attosecond transition absorption spectroscopy in helium atoms, significantly boosting quasi-forbidden transitions. This enhancement technique has potential applications in various fields, including spectral diagnostics of complex molecules and precision spectroscopy of weak transitions in metastable atomic nuclei.

I recommend the article for publication in Communications Physics (authors comment: a slip of the pen) as the claims made by the authors are clearly articulated and supported by both experimental observations and accurate numerical calculations, which are in agreement with one another. As mentioned in the abstract, this topic may be of interest to researchers beyond the specific application to helium.

My only minor concern is that articles in Nature Communications are intended for a broad range of researchers who may not be familiar with the specific types of two-electron levels utilized in this work. While some readers may be comfortable with configurations such as $2p3d$, they may not be as familiar with states such as $sp^2,4^-$. To ensure the article is accessible to a wider audience, it could be beneficial to provide additional explanations or context for these and similar terms.

Otherwise, I support the publication of this article in Nature Communications.

Response: We appreciate the referee's positive assessment of our work and agree with the suggestion that the notation of the doubly excited states should be better explained.

In order to make the whole manuscript accessible to the broad audience of Nature Communications, we added the following sentences to the revised version of our manuscript.

Throughout this work, we followed the early notation of the $^1P^o$ doubly excited states below the $N=2$ threshold of He^+ first introduced by Cooper, Fano, and Prats⁹: $sp_{2,n\pm}$ (the strong mixing of $2snp$ and $2pns$) and $2pnd$, in which the electron configurations are from the shell-model picture (e.g., $2snp$ implies that one electron is in the $2s$ state and the other in the np state). For more classification schemes of these states, see^{13,14}. (Page 4)

Reviewer #3 (Remarks to the Author):

This manuscript describes and demonstrates a very interesting concept for enhancing weak transitions in transient absorption spectroscopy. The idea is to exploit the coupling of the weak transitions to strong transitions by a laser field. The concept is demonstrated on the $2p3d$ and $sp_{2,4-}$ states of helium, the transitions from the ground state to which are nearly forbidden. By coupling them to the strong $2s2p$ transition, the weak transitions are significantly enhanced. Additionally, the intensity of the weak transition is modulated with a period of 1.1 fs, corresponding to the 3.9-eV separation between the $2s2p$ and the $2p3d/sp_{2,4-}$ states.

Overall, this is a very good manuscript presenting convincing experiments and theory. I recommend the following revisions:

Response: We appreciate the referee's positive feedback on our work. Upon the encouraging comments, we implemented the following suggested revisions.

1) *The authors should comment on the time profiles of the absorption line shown in Fig. 3. Whereas the calculations appear to be in line with the mentioned 17-fs decay of the 2s2p state, the experimental absorption signal appears to decay much faster and in a markedly non-exponential fashion, maybe even a time-dependent shift.*

Response: The faster decay of the absorption signal observed in the experiment is attributed to the experimental grazing-incidence geometry, as discussed in our previous paper [V. Stooß *et al.*, Rev. Sci. Instrum. **90**, 053108 (2019)]. The spatial overlap between the XUV and VIS beams at the target medium is optimized at around 0 fs time delay. However, for larger time delays, the movement of the inner mirror leads to a walk-off of the focus of the XUV beam compared to the VIS beam in the interaction region. This reduced beam overlap leads to less effective VIS intensity, resulting in less enhancement of spectral signal at large time delays.

In the revised manuscript, we included comments on this effect: *We note that due to our experimental grazing-incidence geometry¹⁵, the walk-off of the focus of the XUV beam leads to less spatial overlap with the VIS beam in the interaction region at large time delays. This effect gives rise to a faster decay of the experimental absorption signal than the theory results.* (Page 8)

The gentle time-dependent shift shown in the experimental spectrum might be attributed to the Stark shift by the long pedestal of the VIS pulse, and a full knowledge of the pulse parameters in the interaction region would provide more insights on it.

2) *It would be desirable to see a larger portion of the experimental data than shown in Fig. 3a. A presentation of the experimental data as in Ext. Data Fig. 1 would be very helpful to better visualize the experimental results.*

Response: We included the experimental transient absorption spectrum in a large energy range of 58-66 eV in the Supplementary information (Supplementary Figure 3).

Supplementary Figure 3. **Measured transient absorption spectrum in a large energy range.** Contrary to the spectral evolution of $2p3d/sp_{2,4}$ states shown in Fig. 3 in the main manuscript, the spectral amplitude of $2s2p$ state drops significantly in the pulse-overlap region, and gradually rises with increasing time delay. This is related to the loss of amplitude of $2s2p$ state by the action of the coupling VIS pulse when it arrives at the target helium after the XUV pulse. (SI, page 4)

3) *The authors mention the broad applicability of their concept to other types of spectroscopy. The mentioned applications remain somewhat vague, which could be significantly improved by mentioning specific cases to which the present concept could be applied.*

Response: We thank the referee for pointing this out. To make it clearer, in the revised manuscript, we expanded our discussions on the potential applications as follows:

Similar to other well-established methods like cavity-enhanced spectroscopy [Gagliardi & Loock, Springer (2014)] and surface-enhanced infrared spectroscopy [Neubrech et al., Chemical Reviews 117, 5110 (2017)], enhancing single-photon-suppressed transitions in matter can find technological applications in analytical spectroscopy and spectral fingerprinting with enhanced sensitivity in life, chemical, and material sciences. Moreover, with the aid of intermediate states [Martin *et al.*, Phys. Rev. Applied **9**, 014019 (2018)] or bridge processes [Bilous *et al.*, Phys. Rev. Lett. **124**, 192502 (2020)], this concept holds substantial potential for fundamental physics: We envision it to enable identification and

efficient driving of extremely weakly coupled states (including nuclear states) of relevance to future atomic clocks and precision spectroscopy for fundamental-physics tests in searches for physics beyond the standard model¹⁻⁷[Lyu *et al.*, *Commun Phys* **8**, 3 (2025)] (Page 9)

4) Additional details on the experimental parameters should be given to facilitate the reproduction of the data. For example, how were the VIS pulses compressed, how was their duration measured, where was the cutoff of the XUV continuum, etc.

Response: We included the additional details about the experiment in the revised version of the manuscript as a new section (**Experimental apparatus**) in the Methods. The cutoff energy of the XUV continuum was not measured (the XUV spectrum at energies higher than 73 eV is filtered out in the experiment by the Aluminium filter), but a previous streaking measurement yields a representative XUV FWHM duration of ~0.2 fs when the CEP is stabilized [arXiv:2403.02853].

Experimental apparatus

The commercial laser system (FEMTOPOWER™ HE/HR CEP4) delivers 20 fs, 3 mJ pulses with a central wavelength of 780 nm at 3 kHz repetition rate. In order to broaden the spectrum, the laser pulses are guided through a 1.5-m-long double differentially pumped hollow-core fiber, which is filled with helium with a gas pressure of ~2 bar. The pulses are temporally compressed by 7 pairs of chirped mirrors (Ultrafast Innovations PC70) and are characterized by a dispersion-scan (D-scan) setup. The D-scan measurement yields a FWHM pulse duration of 4.6 fs. Afterwards, the few-cycle VIS laser pulses (725-nm central wavelength, ~1-mJ pulse energy) are sent into the vacuum beamline¹⁵ to perform the transient absorption measurements. The XUV pulses are produced by high-harmonic generation (HHG) in a cell containing neon gas with a backing pressure of 100 mbar. To account for the plasma-induced blue shift introduced in the

HHG process, a VIS pulse with a 700-nm central wavelength and 4-fs FWHM duration is employed in the numerical simulation. (Page 10)

5) The numerical parameters used for the calculations shown in Fig. 1 should also be given.

Response: We added the requested parameters in the caption of Fig. 1 accordingly:

Parameters for the calculation results shown in panel **b** are as follows: state $|1\rangle$ ($E_{g1} = 62$ eV, 400-fs decay time); state $|2\rangle$ ($E_{g2} = 60$ eV, 17-fs decay time); transition matrix elements for T_{g1} , T_{g2} , and T_{21} are 0.0008, 0.04, and 2 a.u., respectively; laser 1 [0.2-fs FWHM (full width at half maximum) duration, 1×10^{11} -W/cm² peak intensity, 60-eV central frequency] overlaps temporally with laser 2 (4-fs FWHM duration, 2×10^{12} -W/cm² peak intensity, 700-nm central wavelength). (Page 3)

Overall, this is a very good manuscript, which should be publishable in Nat. Comm. following these revisions.

Referee-Report-Yu-He-NatComm

Title: “Bringing Weak Transitions to Light,” by Yu He et al.

Summary:

A joint theoretical and experimental study of the purely correlated process of double excitation in helium atoms is demonstrated using time-delayed XUV attosecond pulse and visible light. Here, the authors employ a visible light to strongly couple doubly-excited states (DESs) and report on a large enhancement of some weak transitions. For negative time delays, a coherent superposition of several DESs with $^1P^o$ symmetry is created thanks to the broad bandwidth of the weak XUV attopulse centered at 60 eV (near on resonance with the 2s2p DES). Among those states, clearly resolved spectrally are the spectral features of the $sp_{2,n+}$ series and not the 2pnd series and $sp_{2,n-}$ series, as there is a large contrast between their transition probabilities from the ground state. It is only when a few-cycle visible light is present (positive time delays) that the spectral profiles of the $sp_{2,n+}$ series and $sp_{2,n-}$ series are found to be modified. Focusing on the 2p3d and $sp_{2,4-}$ states (separated only by sub-20 meV), with each being strongly coupled to the 2s2p state by two-photon transition via the $2p^2 (^1S^e)$ DES, the authors reported on an enhancement of one order of magnitude for those two states, and that their spectral profile bears close resemblance with window-type resonances.

Results/claims:

The authors report on the first time-resolved observation in the vicinity of the weakly XUV-populated 2p3d and $sp_{2,4-}$ doubly-excited states in helium atoms by ATAS. Although the oscillator strengths of those states are much weaker than that for the 2s2p state, their calculations correctly showed that the transition probabilities from the 2s2p ($^1P^o$) state to the $2p^2 (^1S^e)$ state or from the latter state to the 2p3d/ $sp_{2,4-}$ state are comparable. Therefore, as the energy gap between these states of opposite different symmetry quite matches the wavelength (700 nm) of the tunable visible light, the authors demonstrated that the two-photon coupling between the 2s2p state and each of these two states with the same symmetry mediated by a few-cycle visible light brings those two states (initially buried in noise) to light.

Assessment:

Enhancing single-photon-suppressed transitions, as discussed in this contribution, underlie modern spectroscopy. The concept of introducing another field to induce additional couplings and thus break the scaling law, to enhance some weak transitions is of relevance to optical clocks and fundamental-physics experiments. However, this concept is certainly not new, see for instance [22-24]. Even for helium atoms of interest here, it has already been demonstrated how modified the spectral profiles of strongly coupled states of the $sp_{2,n+}$ series are. The novelty of this work is that it focuses on one state from the 2pnd series and $sp_{2,n-}$ series (whose transition probability from the ground state is much weaker than that for the $sp_{2,n+}$ series) normally buried in noise, and is able to bring them to light (i.e., to enhance their weak transition).

The experimental technique (ATAS) and the theoretical method (TDSE approach within the hyperspherical close-coupling method) employed here are well-established tools in the AMO community. It is only when effects of the spectral broadening and finite spectrometer resolution are included in the theory that theory and experiment agree quite well. Thus, the numerical

method is sound. Technically, the paper is well-structured and clearly written. The figures are carefully designed. The analysis of the results is compelling and supports the conclusions drawn by the authors. However, there are still many aspects regarding the physics and control of this effect that are not elaborated in the main text. I have seven major and some technical issues that the authors should address before I can make a decision on whether the physics discussed in the revised manuscript can be published in Nature Communications.

Major points

1. **Discrepancy between theory and experiment:** In Figure 3, the disagreement between theory and experiment at longer time delays is attributed to the experimental time-delay jitter and spatial beam inhomogeneity. For the absorption profiles at the two representative time delays, why do the optical densities from theory and experiment differ by a factor roughly of two?
2. **Presence of the $2p^2$ ($^1S^e$) state in the absorption profile in Figure 2(c):** The enhancement mechanism proposed the authors involves the $2p^2$ ($^1S^e$) contribution as an intermediate state. If this is correct, then this state should be directly visible in the experimental absorption profile in Fig. 2(c) at lower energy. I urge the authors to show that figure for energy starting from at least 59 eV.
3. **Modification of the $2s2p$ feature:** In the new figure 2(c) above, how does the intensity for the $2s2p$ signal vary when the visible light is on or off? Given the nature of the coupling at that fixed visible pulse intensity, was it expected?
4. **Effects of the visible light intensity:** How do the enhancement of the $2p3d$ and $sp2,4$ -states together with the intensities of the $2s2p$ and $2p^2$ ($^1S^e$) signals change as the visible light intensity increases? What is the minimal and maximal visible intensities for which this effect occurs?
5. **Effects of the visible light CEP:** Given the few-cycle nature of the visible light, can the enhancement be manipulated by varying the carrier-envelope phase of the visible light?
6. **Quantum beats:** Quantum beats between the $2s2s$ and $2p3d/sp2,4$ - states are briefly mentioned on page 8, before Conclusions. Fringes are observed in the upper panels of both Figure 3(a) and 3(b). For both experiment and theory, I urge the authors to display a comparative time-delay dependence of the optical density for three values of the energy: 64 eV, 64.15 eV, 64.2 eV. Take the time delay to be less than the $2s2p$ lifetime.
7. **Intersecting yellowish features in figure 3:** Both theory and experiment in the upper panels of Figure 3 show yellowish intersecting features, which unfortunately are not elaborated in the manuscript. I urge the authors to do so. Are they related to the lifetimes of the doubly-excited states involved in the coupling?

Minor points

1. **Fermi's Golden rule:** The authors claimed in the first paragraph of the introduction that the absorption cross section is proportional to the square of the transition amplitude. Isn't the square modulus? Is the transition matrix element a real or complex number?
2. **Equation (2):** This claim is translated in Equation (2), which would be incorrect if the transition amplitude is complex. In this case, I would expect this equation to be: $|T|^2 + 2\text{Re}[T^*T']$ granted that $|T'|^2$ can be neglected.